# Electrodeposition of a Ni–P–TiO$_2$/Ti$_3$C$_2$T$_x$ Coating with In Situ Grown Nanoparticles TiO$_2$ on Ti$_3$C$_2$T$_x$ Sheets

**Yingchao Du [1,2], Xiaomeng Zhang [1] , Lianqi Wei [1], Bo Yu [1,2], Daqing Ma [3,* and Shufeng Ye [1,*]**

1   State Key Laboratory of Multiphase Complex Systems, Institute of Process Engineering,
    Chinese Academy of Sciences, Beijing 100190, China; ycdu@ipe.ac.cn (Y.D.); xmzhang@ipe.ac.cn (X.Z.);
    lqwei@ipe.ac.cn (L.W.); yubo@ipe.ac.cn (B.Y.)
2   School of Chemical Engineering, University of Chinese Academy of Sciences, Beijing 100049, China
3   China Academy of Safety Science and Technology, Beijing 100012, China
*   Correspondence: madq@chinasafety.ac.cn (D.M.); sfye@ipe.ac.cn (S.Y.);
    Tel.: +86-8491-1220 (D.M.); +86-8254-4899 (S.Y.)

**Abstract:** Protective coatings have received considerable attention for the surface treatment of devices. Herein, in situ grown nanoparticles, TiO$_2$ on Ti$_3$C$_2$T$_x$ sheets (TiO$_2$/Ti$_3$C$_2$T$_x$), are prepared by a simple hydrothermal oxidation method possessing the layer structure, which is applied to prepare protective coatings. The Ni–P–TiO$_2$/Ti$_3$C$_2$T$_x$ coating is prepared by electroplating technology, revealing more excellent properties than those of the Ni–P coating. Compared with the Ni–P coating, even though the Ni–P–TiO$_2$/Ti$_3$C$_2$T$_x$ coating holds the rough surface, the wettability is changed from hydrophilic to hydrophobic, owing to the gathering existence of TiO$_2$/Ti$_3$C$_2$T$_x$ on the surface and coarse surface texture. In addition, the participation of TiO$_2$/Ti$_3$C$_2$T$_x$ in the Ni–P coating can improve the capacity of corrosion prevention and decrease the corrosion rate. According to the results of hardness and wear tests, microhardness of the Ni–P–TiO$_2$/Ti$_3$C$_2$T$_x$ coating is approximately 1350 kg mm$^{-2}$ and the coefficient of friction (COF) of Ni–P–TiO$_2$/Ti$_3$C$_2$T$_x$ coatings is about 0.40, which is much lower than that of Ni–P coatings. Thus, the Ni–P–TiO$_2$/Ti$_3$C$_2$T$_x$ coating can be a promising material to protect the surface of equipment.

**Keywords:** TiO$_2$/Ti$_3$C$_2$T$_x$; Ni–P–TiO$_2$/Ti$_3$C$_2$T$_x$ coatings; microhardness; corrosion; wear

## 1. Introduction

It is well known that the surface is the most essential part of several engineering components, easily suffering from corrosion, abrasion, and fatigue [1,2]. Researchers have made great efforts to improve these surface properties, such as hardness, wear performance, and corrosion resistance [3,4]. Similarly, various technologies have been employed to prepare the protective coating, including electroplating, plasma thermal spray, and chemical vapor deposition [5]. Due to low cost and easy operation, electroplating is the most outstanding among these technologies [6–9].

According to previous papers, the Ni–P composite coating possesses excellent properties, so it is widely used to deal with problems in the engineering process [10,11]. However, when encountering some tricky conditions, the Ni–P composite coating is hardly able to overcome these difficulties [12]. Thus, the modification of Ni–P composite coatings is proven to be an efficient method to expand application fields, so introducing particles into these coatings can improve their properties, such as TiO$_2$ particles, MoS$_2$ particles, and Al$_2$O$_3$ particles [13–16]. Generally, the property of the composite coating is closely related to these incorporating particles [17].

Usually, two-dimensional (2D) structural materials hold a low friction coefficient, so they can show promising prospects in solid lubrication. As a typical layered structural material, $Ti_3C_2T_x$ particles have been applied to lubricating oil, owing to wear resistance, thermal stability, and electrochemical corrosion [18]. Due to the weak Van der Waal's bonding between layers, there is potential for $Ti_3C_2T_x$ particles to slide, so it holds a low friction coefficient of 0.1 in lubricating oil. Thus, these incorporating $Ti_3C_2T_x$ particles can provide a low friction coefficient for the Ni–P coating [19]. As previous papers have shown, $TiO_2$ particles introduced into coatings can increase the hardness, wear resistance, and corrosion resistance. Baghery et al. prepared the Ni–$TiO_2$ coating, which exhibited excellent properties of hardness and wear resistance [2]. Similarly, Uttam and Duchaniya synthesized the Ni–P–$TiO_2$ coating on mild steel, showing excellent corrosion resistance [20]. Therefore, it is a perfect idea for introducing $TiO_2$ particles into the Ni–P coating.

$TiO_2$/$Ti_3C_2T_x$ particles with heterogeneous interfaces have been widely used in photocatalytic activity and electrochemistry. In previous works, Xu et al. prepare in situ grown nanocrystal $TiO_2$ on 2D $Ti_3C_2$ nanosheets for artificial photosynthesis of chemical fuels [21]. Peng et al. found that hybrids of two-dimensional $Ti_3C_2$ and $TiO_2$ exposing {001} facets enhanced photocatalytic activity [22]. However, it is little reported that $TiO_2$/$Ti_3C_2T_x$ particles are used in the area of surface treatment. In this work, in situ grown nanocrystal $TiO_2$ on $Ti_3C_2T_x$ particles were prepared by hydrothermal reaction and incorporated into Ni–P coatings to prepare the Ni–P–$TiO_2$/$Ti_3C_2T_x$ coating. These particles can combine excellent properties with $TiO_2$ and $Ti_3C_2T_x$, so it is beneficial to improve the wear resistance and corrosion resistance of the Ni–P coating.

## 2. Experiment Procedure

### 2.1. Preparation of TiO₂/Ti₃C₂Tₓ Powders

$Ti_3AlC_2$ powders (10 g) were transferred to a Teflon beaker, and 40 mL 40% hydrofluoric acid were dropwise added into them under stirring at 40 °C for 18 h. Then, these obtained products were washed with deionized water until a pH of 5 and dried in a vacuum drying oven at a temperature of 80 °C for 24 h [18].

Those as-prepared products (2.4 g) were added to a beaker, containing 360 mL 1M hydrochloric acid and 7.2 g sodium fluoroborate under agitation for 30 min. The obtained solution was shifted into hydrothermal reactors at 160 °C for 12 h. After washing with deionized water several times, these products were shifted to a vacuum drying oven at 80 °C for 24 h.

### 2.2. Preparation of Composite Coatings

$TiO_2$/$Ti_3C_2T_x$ powders (4 g·L$^{-1}$) were dispersed uniformly into nickel sulfate plating baths, including nickel sulfate, nickel chloride, boric acid, citric acid, and SDS under stirring. The bath compositions and experimental conditions are shown in Table 1. A pure nickel plate (70 mm × 60 mm) and manganese steel (20 mm × 30 mm) were taken as the anode and cathode, respectively. The composition of the manganese steel is shown in Figure 1, mainly containing elements of Fe, Mn, Cr, Si, S, and P. After the process of electroplating, the composite coating was washed by ultrasound for 10 min, dried in an oven at 100 °C for 1 h, and then annealed in a tube furnace at 400 °C for 60 min under argon atmosphere.

### 2.3. Material Characterization

Field-emission scanning electron microscopy (FE-SEM, JEOL JSM-6700F, JEOL, Tokyo, Japan) and a transmission electron microscopy (TEM, JEOL JEM-2010F, JEOL, Tokyo, Japan) equipped with an accelerating voltage of 200 kV were employed to observe the morphology and structure of these samples. X-ray diffractometer (XRD, Philips, X'Pert Pro, Almelo, the Netherlands) with Cu-K$\alpha$ radiation and X-ray fluorescence spectrometer (XRF; AXIOS-MAX, PANalytical B.V., Almelo, the Netherlands) were used to measure the chemical composition of samples. A laser-scanning Raman microscope

(Nanophoton Corporation, Raman-11, Osaka, Japan) was applied to detect the structure of samples with a confocal laser spectrometer, using the 532 nm excitation of the argon laser at room temperature. X-ray photoelectron spectroscopy (XPS, Themo Fisher Scientific, ESCALAB 250Xi, Waltham, MA, USA) with a monochromatic Al Kα radiation was used to record the surface analysis of $TiO_2/Ti_3C_2T_x$ powders. The Fourier transform infrared (FTIR, Excalibur 3100, Varian Medical Systems, Palo Alto, CA, USA) spectroscopy measurement was conducted in a KBr pellet at room temperature. A digital micro-hardness tester (MC010, Yanrui, Shanghai, China) was applied to determine the microhardness of composite coatings at a load of 300 g for 15 s, and the result was achieved from the average of five data on the sample surface. A surface roughness tester (TR211, Shidai Ruida, Beijing, China) was used to detect the Ra of composite coatings, which was obtained from the mean result of five data sets. An electrochemical station (UN-O-16076, Zahner, Kronach, Germany) was used to test the electrochemical measurement in a three-electrode system. Water contact angle measurements using 2 uL water droplets were determined by a measurement apparatus (K100, Kruss, Hamburg, Germany) at 25 °C with DropSnake to shape the drop and measure the contact angle. Wear tests were carried out by a circumrotating ball-on-disk tribometer (HT-1000, Lanzhou, China) with silicon nitride as the grinding material.

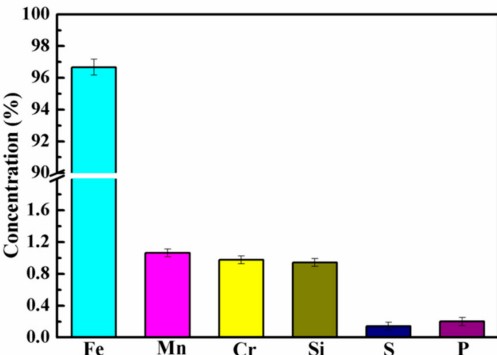

**Figure 1.** X-ray fluorescence (XRF) results of the composition for the manganese steel.

**Table 1.** Bath compositions and experimental conditions.

| Compositions | Experimental Conditions |
| :---: | :---: |
| Nickel sulfate | 250 g $L^{-1}$ |
| Nickel chloride | 40 g $L^{-1}$ |
| Boric acid | 40 g $L^{-1}$ |
| Citric acid | 45 g $L^{-1}$ |
| Sodium hypophosphite | 30 g $L^{-1}$ |
| $TiO_2/Ti_3C_2T_x$ powders | 4 g $L^{-1}$ |
| SDS | 1 g $L^{-1}$ |
| pH | 3–4 |
| Current condition density | 2 A $dm^{-2}$ |
| Time | 1 h |
| Temperature | 50 °C |
| Magnetic stirring speed | 600 rpm |

## 3. Results and Discussions

### 3.1. Characterization of Powders

Figure 2 shows the XRD pattern of $TiO_2/Ti_3C_2T_x$ powders with $TiO_2$ and $Ti_3AlC_2$ standard patterns. Due to no information of $Ti_3C_2T_x$ in ICSD, a $Ti_3AlC_2$ standard pattern with the space group *P63/mmc* was used to fit the XRD pattern of $TiO_2/Ti_3C_2T_x$ powders [23]. These characteristic peaks located at 9.58° and 19.17° were assigned to (002) and (004), which shifted lower to 8.95° and 17.91°,

indicating the existence of $Ti_3C_2T_x$. In addition, the shift of planes can show the increase of the c-lattice parameter and basal spacing. After hydrothermal oxidation, peaks belonging to anatase $TiO_2$ phase were much stronger than those of $Ti_3C_2T_x$ powders, demonstrating that $Ti_3C_2T_x$ was partly oxidized and the anatase $TiO_2$ was present on the $Ti_3C_2T_x$ [22].

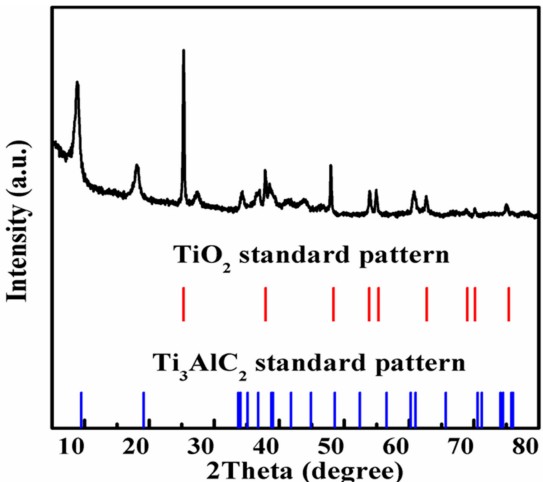

**Figure 2.** X-ray diffraction (XRD) pattern of $TiO_2/Ti_3C_2T_x$ powders with $TiO_2$ and $Ti_3AlC_2$ standard patterns.

Raman spectra and FTIR spectra of $TiO_2/Ti_3C_2T_x$ and $Ti_3C_2T_x$ powders are shown in Figure 3. In Figure 3a, the strongest peak located at 150 cm$^{-1}$ is attributed to the symmetry vibration $E_g$ of anatase $TiO_2$. Similarly, peaks of 413, 520, and 634 cm$^{-1}$ are assigned to the vibration mode of $B_{1g}$, $A_{1g}$, and $E_g$ of anatase $TiO_2$, separately [24,25]. According to the characteristic anatase Raman peak of pure $TiO_2$, a slight increase in the Eg1 peak is detected [26]. The peaks are in a tetragonum in Figure 3, showing the existence of $TiO_2$. These peaks of anatase $TiO_2$ mainly appear among 800~500 cm$^{-1}$, so peaks of $TiO_2/Ti_3C_2T_x$ are stronger than those of $Ti_3C_2T_x$. In addition, the peak located at ~570 cm$^{-1}$ can be assigned to the deformation vibration of the Ti–O bond. [25] Thus, these results of Raman and FTIR spectra indicate the formation of anatase $TiO_2$.

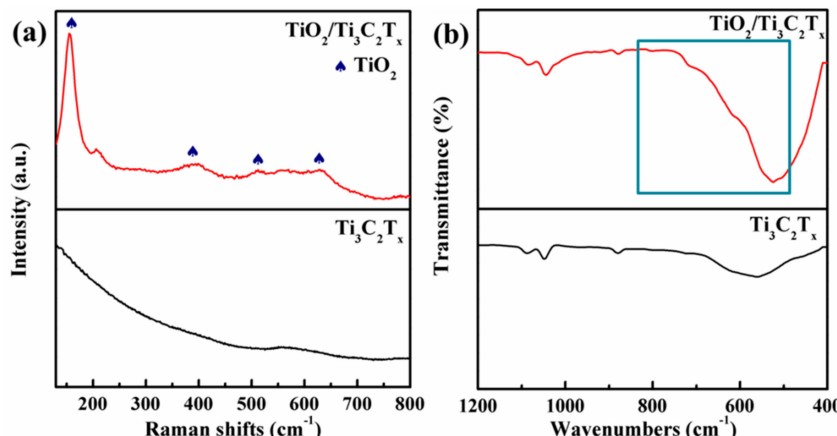

**Figure 3.** (**a**) Raman spectra and (**b**) Fourier transform infrared (FTIR) spectra of $TiO_2/Ti_3C_2T_x$ and $Ti_3C_2T_x$ powders.

SEM images of $Ti_3C_2T_x$ powders and $TiO_2/Ti_3C_2T_x$ powders with EDS elemental mapping are clearly seen in Figure 4. In Figure 4a, the layered structure of $Ti_3C_2T_x$ powders is obviously shown after the etching effect of HF solution. In Figure 4b, after the hydrothermal oxidation, even though anatase

$TiO_2$ particles exist on the $Ti_3C_2T_x$ powders, the layered structure of $Ti_3C_2T_x$ powders still remain, deriving from the titanium atoms on $Ti_3C_2T_x$, acting as nucleating sites for the growth of anatase $TiO_2$ [27]. The HRSEM image of the circle area displays the anatase $TiO_2$ with a thickness of ~30 nm and length of ~200 nm, and it can be observed that the interfacial angle between the {001} and {101} facets of anatase $TiO_2$ is about 68.3° [28]. As for the previous paper, the percentage of the {001} facets can be calculated as about 69.8% from the schematic diagram of an anatase $TiO_2$ [29]. The elemental mapping results from Figure 4c show that main elements of oxygen, titanium, and carbon are present on the $TiO_2/Ti_3C_2T_x$ powders, so it corresponds to results of the XRD pattern (Figure 2).

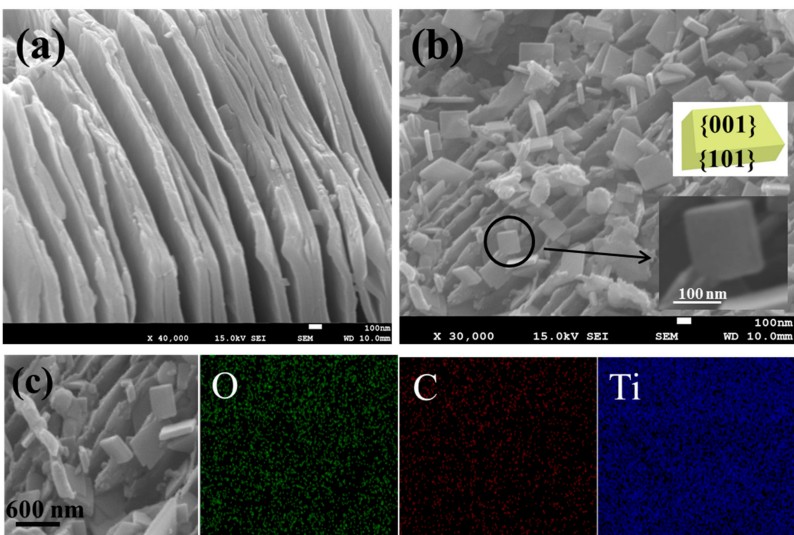

**Figure 4.** Scanning electron microscopy (SEM) images of (**a**) $Ti_3C_2T_x$ powders, (**b**) $TiO_2/Ti_3C_2T_x$ powders with insets of HRSEM image in the area of the black circle and schematic diagram of an anatase $TiO_2$ crystal, and (**c**) $TiO_2/Ti_3C_2T_x$ powders with EDS images.

In Figure 5, HRTEM images show the crystal relationship of $TiO_2$ and $Ti_3C_2T_x$. Figure 5a reveals the morphology of $TiO_2$ with a tetragonal shape, and Figure 5b shows the growth of $TiO_2$ derived from $Ti_3C_2T_x$. In Figure 5c, the interface between $TiO_2$ and $Ti_3C_2T_x$ can be clearly demonstrated by the yellow line. From Figure 5d, the result of FFT demonstrates the existence of anatase $TiO_2$ with crystal planes of (01-1) and (100), while it is clearly seen that the FFT result shows the presence of hexagonal $Ti_3C_2T_x$ with crystal planes of (006) and (103) in Figure 5e. Figure 5f shows that the seamless connection of $TiO_2$ and $Ti_3C_2T_x$ is due to the small discordance between {103} of $Ti_3C_2T_x$ and {11-1} of $TiO_2$ [22]. These results show that $TiO_2$ crystals are present on the crack of $Ti_3C_2T_x$ sheets, indicating that the nucleation of $TiO_2$ may exist at the defective positions of $Ti_3C_2T_x$. In addition, $TiO_2$ at the defect sites of $Ti_3C_2T_x$ may come from hydrated $Ti^{3+}$ ions, which are from the titanium of $Ti_3C_2T_x$ [30,31].

From Figure 6, XPS spectra of Ti2p, C1s and O1s for $Ti_3C_2T_x$, and $TiO_2/Ti_3C_2T_x$ are clearly observed. In Figure 6a, the Ti2p spectra are traced to $Ti2p_{3/2}$ and $Ti2p_{1/2}$ with four pairs and each pair separation of 5.7 eV. These peaks of $Ti2p_{3/2}$ focused on 454.9, 455.8, 456.6, and 459.4 eV are attributed to Ti–C, Ti–X, $Ti_xO_y$, and $TiO_2$, respectively. In addition, the intensity of $TiO_2$ becomes strong and sharp after the hydrothermal oxidation attributed to $TiO_2$ (Ti ions in the valence of $Ti^{4+}$), while other peaks decrease, suggesting the formation of $TiO_2$ from $Ti_3C_2T_x$. As shown in Figure 6b, the C1s is fitted with five peaks, which are located at 281.6, 282.8, 284.8, 286.2, and 288.6 eV, assigned to Ti–C, C–Ti–$O_a$ coming from the adsorbed –OH, C–C, C–O, and C–F, respectively. Due to the appearance of heterojunctions in MXene layers, the intensity of the peak is located at 281.6 eV. The hydrothermal oxidation reduces the appearance of C–Ti–$O_b$, derived from the interface of $TiO_2$ and $Ti_3C_2T_x$ powders [32]. As for the O1s spectra in Figure 6c, four peaks of 529.4, 530.4, 531.8, and 533.3 eV are taken for adsorbed O, Ti–O–Ti, Ti–OH, and C–OH, separately [33]. Thus, this further verifies the presence of $TiO_2$ transformed from $Ti_3C_2T_x$.

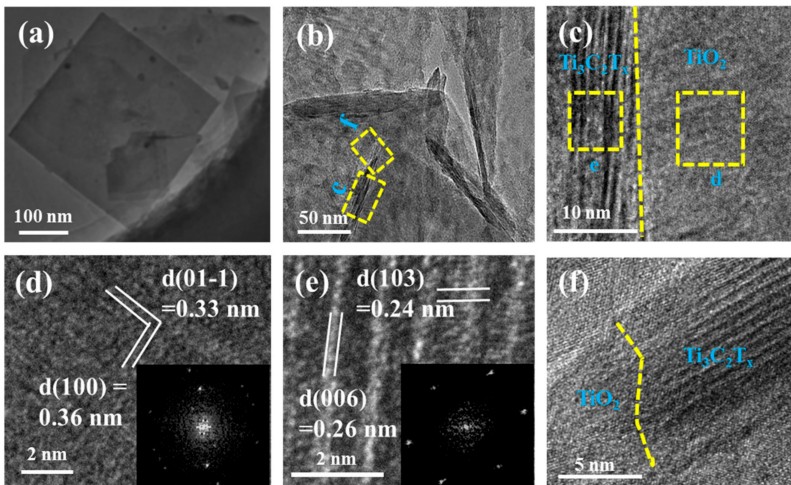

**Figure 5.** (**a**,**b**) TEM image of TiO$_2$/Ti$_3$C$_2$T$_x$; (**c**,**f**) HRTEM images of the yellow gridlines in (**b**); (**d**,**e**) HRTEM images of the yellow gridlines in (**c**) with the insets of FFT.

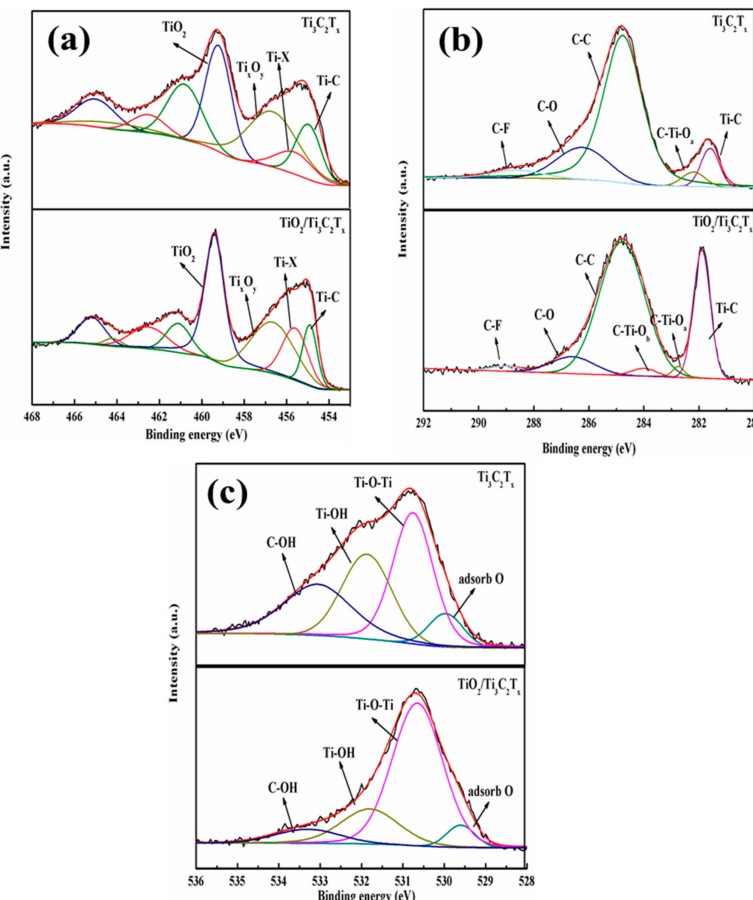

**Figure 6.** (**a**) Ti2p, (**b**) C1s, and (**c**) O1s XPS spectra of Ti$_3$C$_2$T$_x$ and TiO$_2$/Ti$_3$C$_2$T$_x$ powders.

*3.2. Characterization of Composite Coatings*

3.2.1. Compositions of Composite Coatings

XRD patterns of the Ni-P coating and the Ni–P–TiO$_2$/Ti$_3$C$_2$T$_x$ coating are shown in Figure 7. It is clear to see that TiO$_2$ and Ti$_3$C$_2$T$_x$ particles are introduced to the Ni–P–TiO$_2$/Ti$_3$C$_2$T$_x$ coating with these characteristic peaks of 8.95° and 25.292°, respectively. After the calcination, main phases of Ni and

$Ni_3P$ are present in both coatings, but the peak intensity of the Ni phase increases with respect to the $Ni_3P$ phase due to the participation of $TiO_2/Ti_3C_2T_x$ particles. In addition, $TiO_2/Ti_3C_2T_x$ particles can reduce the oriented growth of Ni, which can be determined by the following equation:

$$T = \frac{I_{(hkl)} / I_{0(hkl)}}{\sum_{i=1}^{n} I_{(hkl)} / I_{0(hkl)}} \times 100\% \tag{1}$$

where $I_{(hkl)}$ and $I_{0(hkl)}$ represent the diffraction intensity of the sample and the standard powder, respectively. In addition, n is the number of crystal planes, and T is the indices of crystal direction. As shown in Table 2, the indices of crystal direction for Ni phases in $Ni–P–TiO_2/Ti_3C_2T_x$ coatings demonstrating the T of (111) are relative high, so the oriented growth plane of Ni is (111). Thus, the addition of $TiO_2/Ti_3C_2T_x$ particles has a great effect on the composition of matrix coatings.

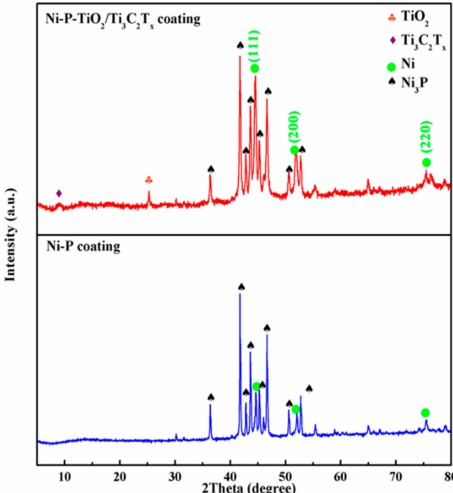

**Figure 7.** XRD patterns of the Ni–P coating and the $Ni–P–TiO_2/Ti_3C_2T_x$ coating.

**Table 2.** Crystal planes and indices of crystal direction in the $Ni–P–TiO_2/Ti_3C_2T_x$ coating.

| Crystal Planes | Indices of Crystal Direction (%) |
|:---:|:---:|
| (111) | 47.17 |
| (200) | 32.39 |
| (220) | 20.44 |

In Figure 8, a Raman spectrum of the $Ni–P–TiO_2/Ti_3C_2T_x$ coating is shown with the image of the diffraction area. From Figure 8a, the addition of $TiO_2/Ti_3C_2T_x$ makes the surface of the $Ni–P–TiO_2/Ti_3C_2T_x$ coating rough, so the $TiO_2/Ti_3C_2T_x$ particles tend to gather together on the surface. In Figure 8b, peaks of 150, 413, 520, and 634 $cm^{-1}$ are derived from $TiO_2$, which corresponds to the result of Figure 3a [24]. In addition, peaks located at 1380 and 1590 $cm^{-1}$ are assigned to D-band and G-band of $Ti_3C_2T_x$ particles. Thus, it further verifies that the $Ni–P–TiO_2/Ti_3C_2T_x$ coating is successfully prepared containing $TiO_2/Ti_3C_2T_x$ particles. In Figure 8c,d, cross-section SEM images show the thickness of Ni–P coatings and $Ni–P–TiO_2/Ti_3C_2T_x$ coatings. According to the contrast of backscattered electron imaging, the boundary between the coating and manganese steel is clear to see, so the thickness of Ni–P and $Ni–P–TiO_2/Ti_3C_2T_x$ coatings is about 10.7 and 16.3 um, respectively.

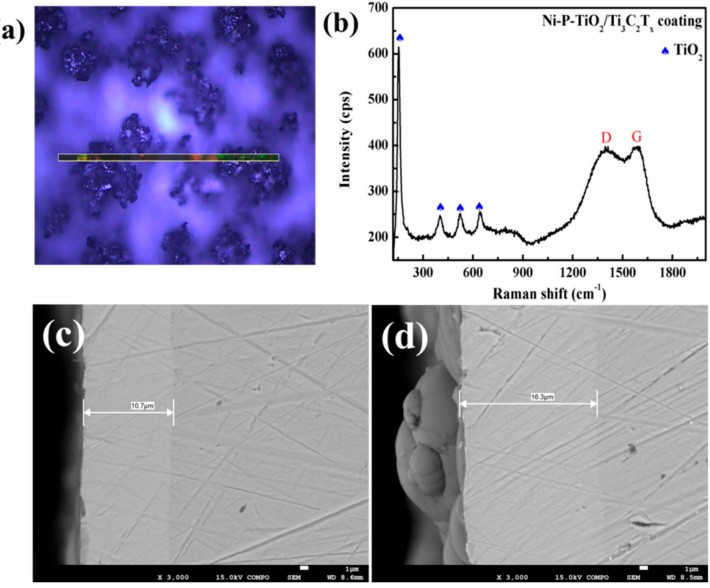

**Figure 8.** (**a**) Image of Raman diffraction area and (**b**) Raman spectrum of Ni–P–TiO$_2$/Ti$_3$C$_2$T$_x$ coatings; (**c**) and (**d**) cross-section SEM images of Ni–P coatings and Ni–P–TiO$_2$/Ti$_3$C$_2$T$_x$ coatings.

As shown in Figure 9, the mechanism of TiO$_2$/Ti$_3$C$_2$T$_x$ particles introduced into Ni–P deposits contains five steps. Cations and surfactants are absorbed into the TiO$_2$/Ti$_3$C$_2$T$_x$ particle to form the clouding in the bulk layer. The clouding of charged TiO$_2$/Ti$_3$C$_2$T$_x$ particles transfers to the cathode through a convection layer and diffusion layer, depending on electrophoresis in large part. The electrical double layer is close to the Ni–P composite coating, which is adsorbed and traps TiO$_2$/Ti$_3$C$_2$T$_x$ particles [6–8]. The formation of the Ni–P coating is an induced codeposition, so reduction of nickel can lead to the reduction of phosphorus, which is beneficial for the embedding and burial of TiO$_2$/Ti$_3$C$_2$T$_x$ particles into the Ni–P coating. Therefore, it is concluded that physical dispersion and electrophoretic migration participate in the formation of the Ni–P–TiO$_2$/Ti$_3$C$_2$T$_x$ coating [9]. After the calcination, Ni and Ni$_3$P phases can be found, which is in accordance with these results of the XRD pattern (Figure 7) and Raman spectrum (Figure 8).

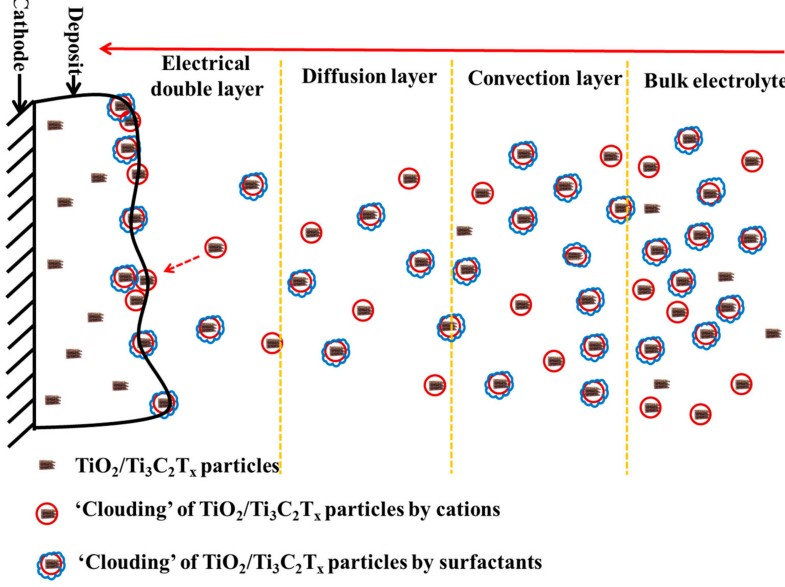

**Figure 9.** Mechanisms of TiO$_2$/Ti$_3$C$_2$T$_x$ particles' codeposition into the Ni–P deposit.

### 3.2.2. Wettability of Composite Coatings

Figure 10 shows SEM images of surface morphology for composite coatings and images of water drops. It is evident that the surface of Ni–P–TiO$_2$/Ti$_3$C$_2$T$_x$ coatings is much rougher than that of Ni–P coatings, owing to the participation of TiO$_2$/Ti$_3$C$_2$T$_x$ gathering on the surface. In addition, the contact angle of Ni–P–TiO$_2$/Ti$_3$C$_2$T$_x$ coatings is larger than that of Ni–P coatings, indicating that the addition of TiO$_2$/Ti$_3$C$_2$T$_x$ can lead to a change of wettability from hydrophilic to hydrophobic. It is well known that surface texture and surface energy can determine the surface wettability, which is important for the corrosion resistance of composite coatings [9]. Therefore, the gathering of TiO$_2$/Ti$_3$C$_2$T$_x$ on the surface and the surface roughness of the composite coating are synergistic actions for the change of wettability. The result of $R_a$ is shown in the SEM image, so it is clear to see that the incorporation of TiO$_2$/Ti$_3$C$_2$T$_x$ can change the surface roughness of the Ni–P composite coating.

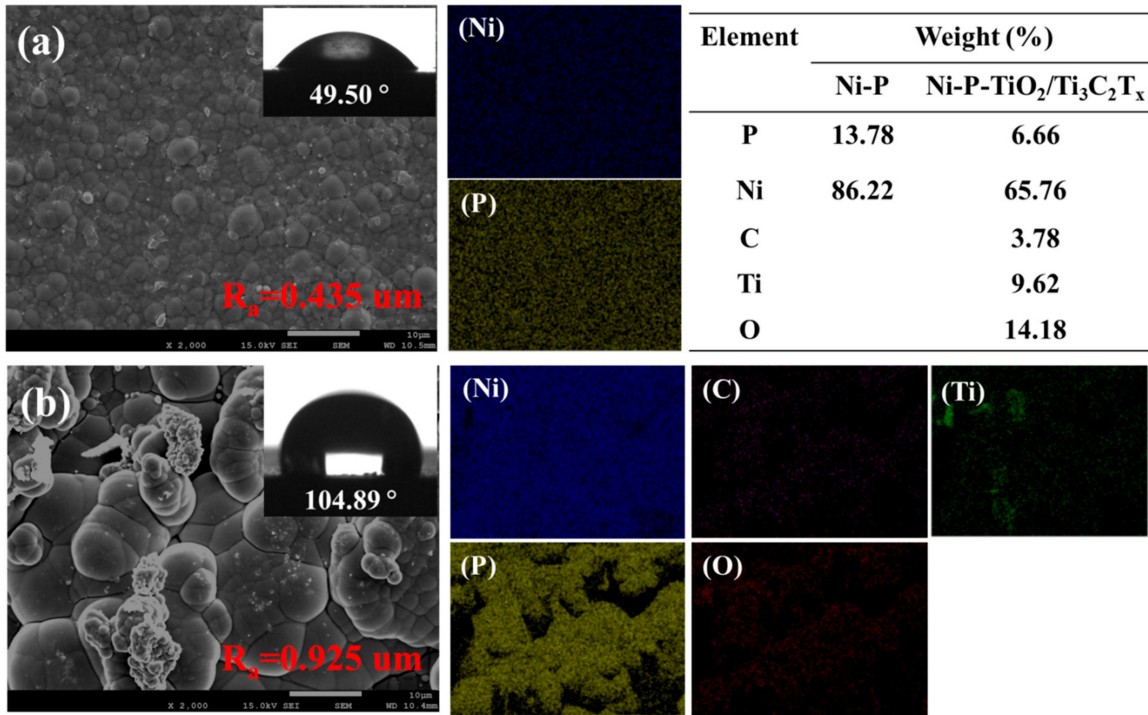

| Element | Weight (%) | |
|---|---|---|
| | Ni-P | Ni-P-TiO$_2$/Ti$_3$C$_2$T$_x$ |
| P | 13.78 | 6.66 |
| Ni | 86.22 | 65.76 |
| C | | 3.78 |
| Ti | | 9.62 |
| O | | 14.18 |

**Figure 10.** SEM images of (**a**) Ni–P coatings and (**b**) Ni–P–TiO$_2$/Ti$_3$C$_2$T$_x$ coatings with the EDS results (insets with images of water drops, the contact angles, and the result of $R_a$).

According to the EDS results, Ni and Ni$_3$P phases are uniformly distributed in the Ni–P coating, while these phases are aggregated partially. In addition, TiO$_2$/Ti$_3$C$_2$T$_x$ particles are present on the surface of Ni–P–TiO$_2$/Ti$_3$C$_2$T$_x$ coatings. Compared with nickel content, the incorporation of TiO$_2$/Ti$_3$C$_2$T$_x$ can decrease the relative content of phosphorus for Ni–P–TiO$_2$/Ti$_3$C$_2$T$_x$ coatings, which is likely to grow more active sites for nickel due to the existence of TiO$_2$/Ti$_3$C$_2$T$_x$ [19].

### 3.2.3. Corrosion Behavior of Composite Coatings

It is shown in Figure 11 that the electrochemical measurement is conducted at room temperature. In Figure 11a, Tafel polarization curves are processed by the Tafel extrapolation method, which can obtain the data of corrosion potential ($E_{corr}$), corrosion current ($i_{corr}$), and polarization resistance ($R_p$). It is obvious that the corrosion current decreases and the corrosion potential transforms to positive potential of the Ni–P coating with the addition of TiO$_2$/Ti$_3$C$_2$T$_x$. In addition, the polarization resistance ($R_p$) of the Ni–P–TiO$_2$/Ti$_3$C$_2$T$_x$ coating is higher than that of the Ni–P coating, confirming the improvement of corrosion protection for the Ni–P–TiO$_2$/Ti$_3$C$_2$T$_x$ coating. As previous papers indicated, TiO$_2$/Ti$_3$C$_2$T$_x$ can hinder the initiation and defect corrosion by regulating and controlling

the microstructure of the composite coating [2]. Furthermore, $TiO_2/Ti_3C_2T_x$ in the Ni–P coating can form corrosion microcells, which serve as the cathode, so it promotes the anode polarization [34–36]. In addition, the passive layer can be seen in the polarization curve of Ni–P–$TiO_2/Ti_3C_2T_x$ coatings, which is good for the corrosion prevention, owing to the gathering of the $TiO_2/Ti_3C_2T_x$ on the surface and microstructure of the surface.

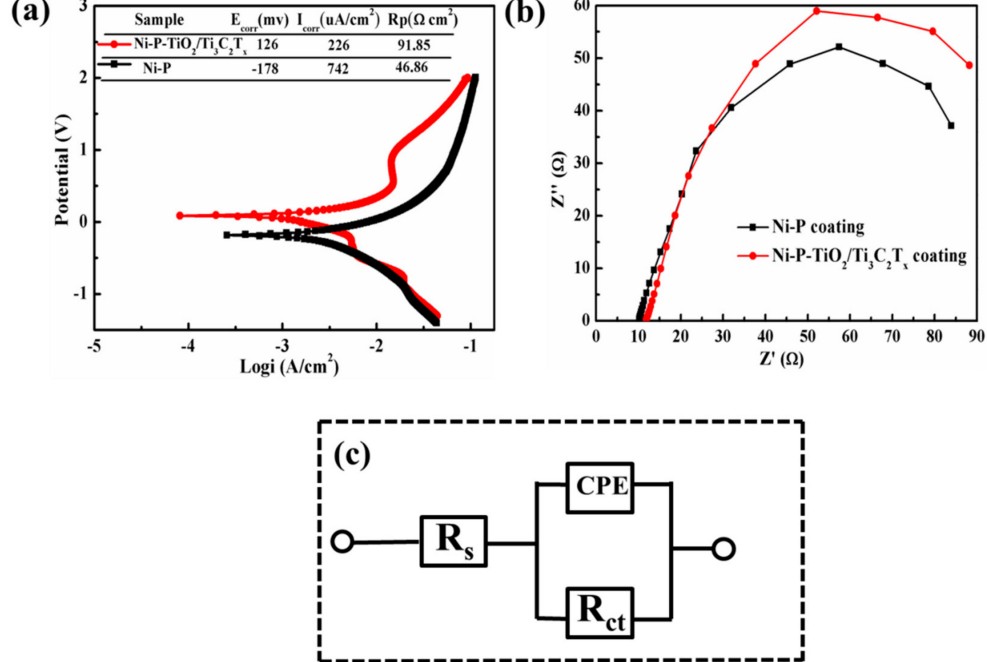

**Figure 11.** (**a**) The polarization curves (inset with the table of the corrosion potential ($E_{corr}$), corrosion current ($i_{corr}$), and corrosion resistance ($R$p); (**b**) nyquist impedance diagrams of Ni–P and Ni–P–$TiO_2/Ti_3C_2T_x$ coatings; (**c**) equivalent circuits used for numerical fitting of impedance plots for 3.5% NaCl solution.

Nyquist impedance diagrams of Ni–P and Ni–P–$TiO_2/Ti_3C_2T_x$ coatings are shown in Figure 11b with the fitting equivalent circle model in Figure 11c, with the average error of 3.5%, which is investigated in the frequency range of $10^{-1}$ to $10^5$. From the result of Figure 11b, it is evident that the electrochemical corrosion is mainly controlled by the diffusion process. In Figure 11c, the equivalent circuit shows the corrosion processes at the electrolyte/coatings interface. $R_s$ stands for the solution resistance, while $R_{ct}$–CPE represents the charge transfer reaction, which is inversely proportional to the corrosion rate. In addition, CPE is a constant phase for a more accurate fit instead of a pure double layer capacitor. $R_{ct}$ stands for the charge transfer resistance, so it is a measure of electron transfer across the surface. Due to the formation of nickel oxyspecies and the attack of $Cl^-$, a charge transfer resistor ($R_{ct}$) is employed to stimulate these reactions [37,38]. In Table 3, these fitting resistances of the equivalent-circuit elements contain $R_s$ and $R_{ct}$. The value of $R_{ct}$ is inversely proportional to corrosion rate. High $R_{ct}$ of the Ni–P–$TiO_2/Ti_3C_2T_x$ coating indicates a large corrosion resistance. From these results, it is evident that the addition of $TiO_2/Ti_3C_2T_x$ can improve wear resistance of the Ni–P coating and decrease the corrosion rate, so it can be concluded that the participation of $TiO_2/Ti_3C_2T_x$ can enhance the property of corrosion prevention.

**Table 3.** The fitting resistances of the equivalent-circuit elements.

| Sample | $R_s$ ($\Omega \cdot cm^2$) | $R_{ct}$ ($\Omega\ cm^2$) |
|---|---|---|
| Ni–P coating | 10.01 | 75.89 |
| Ni–P–$TiO_2/Ti_3C_2T_x$ coating | 13.08 | 87.69 |

### 3.2.4. Wear Behavior of Composite Coatings

From Figure 12, hardness and wear tests were conducted at room temperature. In Figure 12a, the fluctuation of the microhardness belonging to the Ni–P–TiO$_2$/Ti$_3$C$_2$T$_x$ coating is much larger than that of the Ni–P coating, highlighting the uneven surface of the Ni–P–TiO$_2$/Ti$_3$C$_2$T$_x$ composite coating with a high R$_a$ for the gathering of TiO$_2$/Ti$_3$C$_2$T$_x$ particles on the surface of Ni–P composite coatings. In addition, the microhardness of the Ni–P coating is about 680 kg·mm$^{-2}$, while that of the Ni–P–TiO$_2$/Ti$_3$C$_2$T$_x$ coating is approximately 1350 kg·mm$^{-2}$, which is due to the dispersion strengthening and increasing grain boundaries hindering dislocation mobility [39,40]. Thus, the addition of TiO$_2$/Ti$_3$C$_2$T$_x$ can enhance the action of dispersion strengthening and augment grain boundaries of these composite coatings, which is beneficial for the improvement of microhardness for the composite coating.

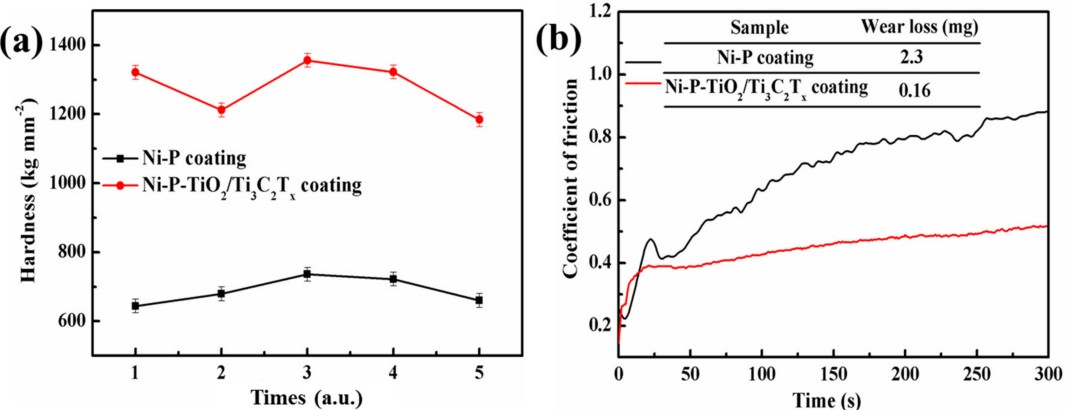

**Figure 12.** (**a**) Microhardness and (**b**) coefficient of friction (inset with the table of wear loss) of Ni–P coatings and Ni–P–TiO$_2$/Ti$_3$C$_2$T$_x$ coatings.

In Figure 12b, results of the wear test are carried out under the condition of dry-grinding, accompanied with a load of 2 N, duration time of 300 s, and circle radius of 1 cm. These composite coatings and the smooth silicon nitride are taken as the stationary disk and the grinding material, respectively. The average coefficient of friction for the Ni–P coating is 0.70 after wear tests, while the average coefficient of friction for the Ni–P–TiO$_2$/Ti$_3$C$_2$T$_x$ coating is down to 0.40 with the participation of TiO$_2$/Ti$_3$C$_2$T$_x$. In addition, the table inserted in Figure 12b illustrates the wear loss after sliding tests, showing that the wear loss of Ni–P coatings and Ni–P–TiO$_2$/Ti$_3$C$_2$T$_x$ coatings is about 2.3 mg and 0.16 mg, respectively. As previous papers have shown, high microhardness and a low coefficient of friction can simultaneously decrease the wear loss of the Ni–P–TiO$_2$/Ti$_3$C$_2$T$_x$ coating [41]. Therefore, the addition of TiO$_2$/Ti$_3$C$_2$T$_x$ to the Ni–P coating can improve the tribological property of this coating.

## 4. Conclusions

In this work, in situ grown nanoparticles TiO$_2$ on Ti$_3$C$_2$T$_x$ sheets were prepared by a simple hydrothermal method. Taking the excellent property of TiO$_2$ and Ti$_3$C$_2$T$_x$ into consideration, TiO$_2$/Ti$_3$C$_2$T$_x$ was incorporated into the Ni–P composite coatings, which were prepared by an electroplating technique, which is little reported in this area. The following conclusions could be obtained from this study:

- The in situ grown nanocrystals TiO$_2$ on Ti$_3$C$_2$T$_x$ sheets can still maintain a layered structure. The Ni–P–TiO$_2$/Ti$_3$C$_2$T$_x$ coating is successfully prepared by an electroplating technique;
- The participation of TiO$_2$/Ti$_3$C$_2$T$_x$ can change the wettability of Ni–P composite coatings from hydrophilic to hydrophobic;
- The Ni–P–TiO$_2$/Ti$_3$C$_2$T$_x$ coating shows better properties of corrosion prevention than Ni–P coatings.

- Microhardness of the Ni–P–TiO$_2$/Ti$_3$C$_2$T$_x$ coating is approximately 1350 kg mm$^{-2}$, and the coefficient of friction of Ni–P–TiO$_2$/Ti$_3$C$_2$T$_x$ coatings is about 0.40, which is lower than that of the Ni–P coating.

**Author Contributions:** Conceptualization, Y.D., D.M., and S.Y.; Formal analysis, Y.D.; Funding acquisition, D.M. and S.Y.; Investigation, Y.D., X.Z., B.Y., and L.W.; Methodology, Y.D., X.Z., and L.W.; Supervision, D.M., Y.D., and S.Y.; Validation, Y.D. and D.M.; Writing—original draft, Y.D.; Writing—review and editing, Y.D. and S.Y.

**Funding:** We are thankful for the "Strategic Priority Research Program" of the Chinese Academy of Sciences (grant number XDA09040102) and National Key R and D Program of China (grant number 2016YFC0801500) for supporting this work.

**Conflicts of Interest:** The authors declare no conflict of interest.

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
