# Peer review of "Electrodeposition of a Ni–P–TiO2/Ti3C2Tx Coating with In Situ Grown Nanoparticles TiO2 on Ti3C2Tx Sheets"

_coatings, doi:10.3390/coatings9110750_

Round 1

Reviewer 1 Report

need more detailed analysis for the corrosive mechanism (p.246) and the wear mechanism (p.271)

Reviewer 2 Report

Paper can be accepted after the following corrections:

Possible applications should be precisely and clearly stated in Introduction. Please don't start the section from a number (e.g. lines 62, 66, 71). Figure 6 (all sub-figures) should be enlarged to clarify the figure. Figure 9 is not clear. Please prepare it accordingly to the scientific standards. Sub-figure 11c should be explained in the figure caption.

Reviewer 3 Report

The authors provide a paper dealing with the electrodeposition of complex Ni-P-TiO2/Ti3C2Tx coatings. The paper can be of interest for Coatings but MAJOR revisions are requested.

I would simplify the text because it difficult to understand with a lot of chemical formulas. I would like that the authors discuss in a detailed way the role of film stoichiometry during the nanoparticles growth and deposition. This I believe is of extremely importance. As a matter of facts, TiO2 can be non-stoichiometric affecting the properties of the coating. The authors can comment on the basis of the following paper doi.org/10.1016/j.matdes.2018.06.051 in which the (optical) of TiO2 are discussed especially focusing on composition effects. Please also comment on that when performing the Raman analysis in which it can be calraly detected the stoichiometry of the TiO2. I found strange the indentation analysis. First of all, I would suggest to the authors to comment about the effect of film surface (roughness) on the mechanical properties (Figures 4-10) then I would prefer that the authors carry out NANOindentation maybe in Continuous stiffness measurement configuration (CSM) to get precise information about H. Fig.12 must be changed providing H in GPa and it is also not clear what is the frequency. Overall, the authors must improve all this part. Concerning the previous analysis, the authors must be aware that there exist some nanoindentation techniques that can provide key information about residual stress. Specifically, I refer to the following paper doi.org/10.1016/j.matdes.2016.06.003 in which the residual stress can be extracted by nanoindentation. I would like that the authors comment on that. The authors must make an overall effort to improve the quality of their work in terms of clarity and English.

Round 2

Reviewer 3 Report

The authors have slightly improved the quality of the paper, but it does not reach the quality to be accepted. Actually, the text is almost unchanged and the response letter is very short.

I would suggest to expand this analysis commenting in a more detailed way (also in the text) about the TiO2 stoichiomety. Then, the main point now is to improve the nanoindentation analysis.

Specifically, I asked the authors to perform nanoindentation in Continuous stiffness measurement (CSM) mode. This would allow to track the H and E as a function of the indentation depth while evaluating the effect of the surface. Please improve this part. Finally, in the bibliography, the authors extend the First Names, while it should be the Family Name. 

Author Response

Thank you very much for your letter and for the Editors’ and Reviewers’ comments concerning our manuscript entitled “Electrodeposition of a Ni-P-TiO2/Ti3C2Tx coating with in-situ grown nanoparticles TiO2 on Ti3C2Tx sheets”. Those comments are all valuable and very helpful for revising and improving our paper, as well as the important guiding significance to our researches. We have studied comments carefully and have made correction which we hope meet with approval. And then, we carry out a comprehensive revision of the manuscript. Revised portion are marked in red in the paper. The revision is addressed below point by point.

Responses to the Editorial comments:

I would suggest to expand this analysis commenting in a more detailed way (also in the text) about the TiO2

Answer: Thank you very much for the comments. Some analysis about TiO2 stoichiomety has been added into the part of 3.1 with more references.

Then, the main point now is to improve the nanoindentation analysis. Specifically, I asked the authors to perform nanoindentation in Continuous stiffness measurement (CSM) mode. This would allow to track the H and E as a function of the indentation depth while evaluating the effect of the surface. Please improve this part.

Answer: Thank you very much for the comments. At first, I am grateful for the comment of the nanoindentation analysis and I have carefully studied the paper (Determination of the elasticmoduli and residual stresses of freestanding Au-TiW bilayer thin films by nanoindentation). The residual stress is important to evaluate the film surface, but it is difficult to perform this test for absence of this apparatus. Then, I am sorry to not perform this test. This paper is about the corrosion and wear property, so I just do tests of microhardness and friction coefficient and not concern the surface residual stress. At last, I am thankful again for learning a lot of knowledge from the comment. 

Finally, in the bibliography, the authors extend the First Names, while it should be the Family Name.

Answer: Thank you very much for the comments. The formation of the bibliography has been rewritten about Reference 5 and 13.

Special thanks to you for your good comments.

Round 3

Reviewer 3 Report

-